# New Bacteriophages with Podoviridal Morphotypes Active against *Yersinia pestis*: Characterization and Application Potential

**DOI:** 10.3390/v15071484

**Published:** 2023-06-30

**Authors:** Tamar Suladze, Ekaterine Jaiani, Marina Darsavelidze, Maia Elizbarashvili, Olivier Gorge, Ia Kusradze, Tamar Kokashvili, Nino Lashkhi, George Tsertsvadze, Nino Janelidze, Svetlana Chubinidze, Marina Grdzelidze, Shota Tsanava, Eric Valade, Marina Tediashvili

**Affiliations:** 1George Eliava Institute of Bacteriophages, Microbiology and Virology (Eliava IBMV), 3, Gotua Str., 0160 Tbilisi, Georgia; tamuna_suladze@pha.ge (T.S.); jaianieka@yahoo.com (E.J.); m.darsavelidze@pha.ge (M.D.); m.elizbarashvili@pha.ge (M.E.); iakusradze@pha.ge (I.K.); t.kokashvili@pha.ge (T.K.); nlashkhi20@gmail.com (N.L.); giatserts@yahoo.com (G.T.); n.janelidze@pha.ge (N.J.); 2French Armed Forces Biomedical Research Institute (IRBA), 1, Place du Général Valérie André—BP 73, 91223 Bretigny-sur-Orge, France; eric.valade@def.gouv.fr; 3School of Science and Technology, University of Georgia, 77a, Kostava Str., 0171 Tbilisi, Georgia; 4National Center for Disease Control and Pubic Health (NCDC), 99, Kakheti Highway, 0109 Tbilisi, Georgia; s.chubinidze@ncdc.ge (S.C.); m.grdzelidze@ncdc.ge (M.G.); sh.tsanava@ncdc.ge (S.T.)

**Keywords:** plague, bacteriophage, phage genome, phage typing, phage therapy

## Abstract

Phages of highly pathogenic bacteria represent an area of growing interest for bacterial detection and identification and subspecies typing, as well as for phage therapy and environmental decontamination. Eight new phages—YpEc56, YpEc56D, YpEc57, YpEe58, YpEc1, YpEc2, YpEc11, and YpYeO9—expressing lytic activity towards *Yersinia pestis* revealed a virion morphology consistent with the *Podoviridae* morphotype. These phages lyse all 68 strains from 2 different sets of *Y. pestis* isolates, thus limiting their potential application for subtyping of *Y. pestis* strains but making them rather promising in terms of infection control. Two phages—YpYeO9 and YpEc11—were selected for detailed studies based on their source of isolation and lytic cross activity towards other *Enterobacteriaceae*. The full genome sequencing demonstrated the virulent nature of new phages. Phage YpYeO9 was identified as a member of the *Teseptimavirus* genus and YpEc11 was identified as a member of the *Helsettvirus* genus, thereby representing new species. A bacterial challenge assay in liquid microcosm with a YpYeO9/YpEc11 phage mixture showed elimination of *Y. pestis* EV76 during 4 h at a P/B ratio of 1000:1. These results, in combination with high lysis stability results of phages in liquid culture, the low frequency of formation of phage resistant mutants, and their viability under different physical–chemical factors indicate their potential for their practical use as an antibacterial mean.

## 1. Introduction

Infectious diseases caused by highly pathogenic bacteria are still a challenge for modern medicine. Relevant infections may occur due to natural outbreaks or through a deliberate release during bio terroristic attacks by biological warfare agents.

The causative agent of severe acute disease plague is *Yersinia pestis*, a non-spore forming, rod-shaped, and facultatively anaerobic microorganism in the family *Enterobacteriaceae,* which is classified as a Class A select agent [1]. Only one serotype of *Y. pestis* is recognized, while pathogenic strains can be divided into four biovars: Antiqua, Medievalis, Orientalis, and the recently described biovar Microtus. *Y. pestis* has three plasmids and several virulence factors, including F1, Murine exotoxin, LPS endotoxin, coagulase, pesticin, and plasminogen activator [2,3,4,5]. The disease caused by *Y. pestis* is highly contagious with a short incubation period and infectivity in low doses, manifested mainly as a bubonic plague with a case-fatality ratio of 30% to 60% that can be developed to a more severe form of a pneumonic plague that is always fatal if left untreated. Due to its potential of person-to-person dissemination via aerosols and high lethality, *Y. pestis* can be used as a warfare agent in aerosolized form [2].

Historically, the plague has caused several pandemics which have led to large numbers of deaths. The plague was categorized as a re-emerging disease because of its reappearance in several countries in the 1990s and reported human plague outbreaks, primarily in Madagascar [6]. The plague also occurred in the past on the territory of Georgia where two natural foci of the plague are distinguished: plain foothills and high mountainous. A total of 46 strains of *Y. pestis* were isolated during the 1979–1997 period in these areas, while no new isolates have been obtained in recent years [3,4,7].

Early diagnostics of the disease and treatment with antibiotics (streptomycin doxycycline and fluoroquinolones) and supportive therapy are believed to be quite effective in treating the plague, although the rapid identification of *Y. pestis* in patients with severe plague, especially those with the aerosolized and dispersed form, and their treatment is still a challenge, requiring parenteral antibiotics. [6,8,9,10,11]. In the last two decades, the problem has been aggravated by the rise of multidrug-resistant strains of *Y. pestis*. In 1995, a strain isolated in Madagascar exhibited resistance to first-rank drugs currently used for plague treatment and prevention [12,13,14,15]. No plague vaccine for public use is currently available. The prophylaxis of medical personnel and scientists is also problematic in case of a plague outbreak. The US vaccine was discontinued in 1999, and besides, it was not protective against the pneumonic plague. New vaccine(s) are under development [8,16,17].

All of the abovementioned indicate the necessity of the development and use of alternative means for treatment and infection control of the plague. Interest in bacterial viruses (bacteriophages) in general as well as to *Y. pestis* phages in particular has been renewed in recent years. In the 1920s, at the beginning of the phage therapy era, Felix d’Herelle used phage therapy against cholera and the bubonic plague [18,19,20,21,22,23]. Considering the increasing incidence of antibiotic resistance of bacterial pathogens and the lack of novel antibiotics, phages attracted serious attention as alternatives to antibiotics [21,24,25,26]. Phage products are stable, safe, and considerably easier to produce and can be regularly modified in response to changes in the susceptibility of target pathogens [27]. Bacteriophages can be used as tools for bacterial detection and identification, subspecies typing, phage therapy prophylaxis, and environmental decontamination [10,28,29,30,31,32,33,34,35,36,37]. All these directions were intensively developed in Georgia at the G. Eliava Institute of Bacteriophages, Microbiology, and Virology (Eliava IBMV) [24,38,39]. In addition to the elaboration and practical application of therapeutic and prophylactic bacteriophages, the phage typing method was used for epidemiological diagnostics and tracking of different medically important bacterial infections [40,41]. Efficient phage typing schemes were elaborated by the Eliava scientists for *Salmonella Typhimurium* and *Pseudomonas aeruginosa* [42,43].

A number of studies targeting phages lysing *Y. pestis* were performed at different scientific centers worldwide. Out of three well-studied and widely used plague diagnostic phages, two, namely ϕA1122 and the Pokrovskaya phage, are similar to *E. coli* phage T7, display high lytic activities, and display a very broad lytic spectrum towards *Y. pestis* strains of different origins, but also express activity to some strains of its closest phylogenetic relative: *Yersinia pseudotuberculosis*. The US CDC recommends phage ϕA1122 as an important diagnostic tool for the plague [10,44,45,46,47]. Another diagnostic phage-L-413C, isolated from lysogenic *Y. pestis* strain 413 (biovar Medievalis), is much more specific. It is active only against *Y. pestis* (as shown on 6000 global isolates) and very rare strains *Escherichia coli* [10,48]. A combination of L-413C and φA1122 phages was shown to be highly effective for the detection and identification of *Y. pestis* by using qPCR with primers specific for phage DNA [10]. Five phages isolated at the Eliava IBMV on other hosts (*Shigella sonnei*, *S. Typhimurium,* and *Y. enterocolotica)* expressed lytic activity towards *Y. pestis* and revealed three distinct phage subtypes based on DNA restriction profiles [49].

The genome analyses of a number of lytic and temperate phages active against *Y. pestis* were reported [50,51,52,53]. Among them, the bacteriophages Berlin, Yepe2, Yep-phi, YpP-R, YpP-G, YpsP-G, and ΦA1122 were shown to be of the *Podoviridae* morphotype. The phages with this morphology have considerably small and conserved genomes, are easy to propagate, and have short lytic cycles which make them promising for therapeutic applications. The lytic bacteriophage PY100 with the *Myoviridae* morphology has a broad host range within the genus *Yersinia*. Another genetically studied phage, L-413C, was shown to be a temperate *Y. pestis* phage. Phage Yep-phi is routinely used as a diagnostic agent for *Y. pestis* in China. The novel *Y. pestis* phage JC221 with the *Myoviridae* morphology has important reference values for the study of environmental microecology and epidemiology of plague foci [52]. The experimental *Y. pestis* phage mixture YPP-100 was shown to be effective for surface decontamination [37]. However, the knowledge about the biocontrol potential and therapeutic importance of *Y. pestis* phages has still to be extended.

The goal of the presented study was to isolate, characterize, and identify naturally occurring bacteriophages active against *Y. pestis* and to evaluate the potential of their application for plague epidemiology (strain subtyping) and for infection control.

## 2. Materials and Methods

### 2.1. Microbiological Media and Buffers

Different nutrient media and solutions were used: Brain Heart Infusion (BHI) broth and agar (Liofilchem, Roseto degli Abruzzi, Italy); Cefsulodin–Irgasan–Novobiocin (CIN) agar (Liofilchem, Roseto degli Abruzzi, Italy); Luria–Bertani (LB) agar and broth (Liofilchem, Roseto degli Abruzzi, Italy); SM buffer containing 5.8 g NaCl, 2.0 g MgSO_4_·7H_2_O, 50 mL 1 M Tris-HCl, in 1 L dH_2_O, pH 7.4; PBS—phosphate-buffered saline, pH 7.4 (Sigma-Aldrich, St. Louis, MO, USA); and M9 minimal medium standard (Sigma-Aldrich).

### 2.2. Bacterial Strains

The following bacterial strains were used in this study: the vaccine strain *Y. pestis* EV76 (the collection of Eliava IBMV) as a standard/reference strain; 35 strains from the collection of French Armed Forces Biomedical Research Institute (IRBA, Bretigny sur Orge, France), including 2 strains of *Y. pseudotuberculosis*, 1 strain of *Y. frederiksenii,* and 32 strains of *Y. pestis* of worldwide origin, among them 10 NCTC strains isolated from different human and animal sources and 22 strains representative of diversity withinthe *Y. pestis* species and mostly isolated on humans in the 20th century (Appendix A); a set of 36 *Y. pestis* strains from a collection of the National Center for Disease Control and Public Health (NCDC, Tbilisi, Georgia), including 9 strains isolated from rodents *Microtus arvalis* and 27 from fleas (20 strains from *Ctenophthalmus teres* and 7 from *Callopsylla caspius*)collected in Georgia in the Ninotsminda region during the period of 1979–1997 (Appendix A). For propagation of the *Y. pestis* strains, BHI broth and agar and selective CIN agar were used. For isolation of phages and phage susceptibility studies, human isolates of different genera and species of the *Enterobacteriaceae* family were used: 4 strains of *Salmonella enteritidis*, 6 of *Salmonella Typhimurium*, 4 of *Salmonella Agona*, 1 strain of each of *Salmonella Paratyphi A* and *Salmonella Oranienburg*, 5 strains of *Shigella flexneri*, 4 of *Shigella sonnei*, 1 of *Shigella boydii*, 21 of *Escherichia coli,* and 45 strains of *Yersinia enterocolitica* (O:3, O:5, O:6, O:8, and O:9 serotypes) (Appendix A). *Enterobacteriaceae* strains were propagated on LB and BHI broth and agar. Strains of *Y. pestis* and *Y. enterocolitica* were grown at 28 °C and other species of *Enterobacteriaceae* were grown at 37 °C.

The work on *Y. pestis* strains was conducted in the conditions of Biosafety Level 3 (in the labs of IRBA, France, and NCDC, Georgia) while the laboratory experiments involving strains of different *Enterobacteriaceae* species and also vaccine strain *Y. pestis* EV76 were conducted in the BSL2 and BSL2+ labs (at the Eliava Institute of Bacteriophages, Georgia).

### 2.3. Bacteriophages

Seventy phages from the Eliava IBMV collection specific to different enteric bacteria (*Salmonella* spp., *Shigella* spp., *E. coli*, and *Y. enterocolitica*) were used along with 54 primary phage isolates active against the *Enterobateriaceae* family obtained during this study, from which 8 phages were consequently propagated on the *Y. pestis* EV76 strain. For propagation and testing of phages, LB and BHI media were used.

### 2.4. Isolation and Primary Detection of Phages from Environmental Samples

A standard enrichment technique was used for the isolation of new bacteriophages from different environmental samples collected from city effluents, soil, rivers, and lakes located in different regions of Georgia [54]. Briefly, a mixture of water or soil suspension (100 mL), appropriate overnight bacterial broth culture (1 mL), and 10 × BHI broth were incubated at 28 °C overnight. After centrifugation and filtration through 0.45 μm cellulose acetate membrane filters, the suspension was spotted on a streak of indicator bacteria on a 1.5% BHI agar plate. After drying of the phage drops, Petri plates were incubated at 28 °C for 24–48 h. The results were assessed by the development of lytic reaction zones registered as CL, complete lysis; SCL, semi-confluent lysis; OL, overgrown lysis in the presence of single bacterial colonies on the spot; IPO/IPC, multiple opaque or clear phage plaques; and R, resistant.

### 2.5. Purification and Propagation of Phages

The soft agar overlay method was used for the enumeration of bacteriophages and to obtain pure phage lines [55]. Briefly, single phage plaques with different morphologies were picked up and transferred into 1 mL BHI broth containing 0.04% chloroform. The obtained suspension was kept at room temperature for 30 min and after plating of its serial 10-fold dilutions in BHI broth, new phage plaques were obtained on BHI agar plates. The purification of phages was repeated three to five times until homogeneous plaques were obtained for each phage isolate. Petri plates with semi-confluent lysis were overlaid with 3 mL BHI broth and the upper 0.5% agar layer was collected, centrifuged at 5000× *g*, and subsequently filtrated through 0.45 µm cellulose acetate membrane filters. The concentration of obtained phage stock was measured by the soft agar overlay method [54,56].

### 2.6. Characterization of Bacteriophages Lytic to Y. pestis

#### 2.6.1. Phage Virion Morphology

The phage nucleocapsid morphology was studied by transmission electron microscopy (TEM) as described in [57] with some modifications. Samples, namely phage lysates in titers of 1 × 10^10^ PFU/mL previously treated with distilled water during 24 h, were prepared on Formvar/carbon film coated 300 mesh copper grids FCF300-CU (Electron Microscopy Sciences, Fort Washington, PA, USA), negatively contrasted with 2% uranyl acetate, and examined in the JEM SX100 (Jeol, Kotyo, Japan) at 80 KV and instrumental magnification 40,000×. Phage particles were measured and the average size of the head and tail were calculated.

#### 2.6.2. Phage Host Range

The phage host range was determined by screening phages against strains of target species using the phage spot test technique. Briefly, the lines of tested strains were performed on 1.5% of BHI agar plates by striking 10 µL of bacterial suspension (1 × 10^8^ CFU/mL). After drying, the bacterial lines were spotted with 10 μL of each phage (1 × 10^7^ PFU/mL) and incubated at 28 °C for 24 h. The lysis intensity was registered as mentioned in Section 2.4. The efficiency of plating (EOP) was calculated by dividing the phage titer on the test strain by the titer of the same phage on its host strain. The phage titer was determined by a soft agar overlay method [54].

#### 2.6.3. Lysis Stability of Phages in Liquid Culture

The lysis stability of phages was studied in liquid media on the host strain *Y. pestis EV76* according to the method of Appelmans [58]. Briefly, serial 10-fold dilutions of phages with initial titers of 2 × 10^10^ PFU/mL were prepared in tubes with 4.5 mL of BHI broth and to each dilution, 100 μL of host strain equal to 0.5 MFTS (McFarland Standards Kit, bioMerieux, Marcy-l’Étoile, France) was added. The multiplicity of infection (MOI) in the tubes ranged from 1000 to 0.0001. The tubes with phage and bacteria were incubated at 28 °C. The phage lytic activity was visually checked at 24 h and 48 h as well as after one week by examining bacterial growth and comparing with MFTS tubes.

#### 2.6.4. Viability of Phages in Different Solutions

The stability of phages was determined in different media and solutions, such as BHI broth, SM buffer, and PBS. The phages were inoculated in test solutions at the final concentration of 2 × 10^8^ PFU/mL and kept at 4 °C in dark conditions with periodic (24 h, 1 week, 2 weeks, 1 month, 6 months, and 1 year) checks of phage viability by the agar overlay method [54].

#### 2.6.5. The Frequency of Formation of Phage-Resistant Bacterial Mutants

The phage suspension with a titer of 10^8^ PFU/mL in the amount of 0.2 mL of was equally distributed on 1.5% BHI agar plates. When the liquid was dried, 0.1 mL of the bacterial suspension in the concentration 10^7^ CFU/mL was applied to the plate and, after 18–24 h of incubation at 28 °C, the number of formed phage-resistant bacterial mutants was recorded. The frequency of mutation was calculated according to the formula: a = Kr/N, where a—the mutation frequency, K—a constant number and equal to 0.3, r—the number of generated phage-resistant mutants, and N—initial number of bacterial cells used in the experiment [59].

#### 2.6.6. Phage Adsorption and One-Step Growth Cycle

The phage one-step growth cycle was studied by a standard methodology [54,56] with some modifications. Adsorption was estimated by counting the numbers of non-adsorbed phage particles. Briefly, 0.9 mL of bacterial culture (1 × 10^8^ CFU/mL) was mixed with 0.1 mL of phage (1 × 10 ^7^ PFU/mL). BHI broth mixed with the same amount of phage was used as a phage control. Both tubes were incubated at 28 °C in a water bath. At certain intervals (0, 2, 4, 6, 8, 10, 12, 14, 16, 18, and 20 min), 0.1 mL samples from the phage–bacteria mixture were diluted in 9.9 mL cold BHI broth with 0.4 mL chloroform and afterwards were kept for 5 min on ice and the number of free phage particles was estimated by the double layer method. The same procedures were conducted for phage control after 15 min. After incubation of the plates at 28 °C for 24 h, non-adsorbed phage plaques were counted. The percent of adsorbed phages was calculated with the following equation: 100 − (P_n_\P_0_ × 100), where P_n_ is the number PFU of non-adsorbed phages and P_0_ is the number of PFU on the control plate.

For estimation of the phage latent period and burst size after completion of the predetermined period of adsorption, the samples were taken at 5 min intervals and the number of phages in the test suspensions was determined by the double-layer agar method. The obtained results were used to determine the latent period and burst size calculated according to the following formula: (P_n_2) × 100/(P_n_1) − non-adsorbed phages/5), where P_n_1 is a diluted mixture of tested suspension in titer 2 × 10^3^ PFU/mL and P_n_2 is a 100 times diluted mixture of P_n_1.

#### 2.6.7. Phage DNA Extraction

For the isolation of DNA from bacteriophages, the concentrated phage suspensions with the titer 1 × 10^10^ PFU/mL were used. DNA was extracted using QIAamp DNA Mini kit (QIAGEN, Hilden, Germany) according to the manufacturer’s instructions. The DNA concentration was measured on a microvolume spectrophotometer Nano Drop One (Thermo Scientific, Waltham, MA, USA).

#### 2.6.8. Full Genome Sequencing of Phage DNA and Analysis

Libraries were prepared for Illumina sequencing with the NEBNext UltraII DNA kit (New England Biolabs, Ipswich, MA, USA) after shearing with the M220 Covaris device (Covaris, Woburn, MA, USA) accordingly to the manufacturer’s recommendations. After size selection with AMPure beads (Beckman, Brea, CA, USA), sequencing was achieved using an HT 2 × 150 bp cartridge on a NextSeq 550 instrument (Illumina, San Diego, CA, USA). Samples’ fastq files were checked for quality by FastQC and reads were de novo assembled using spades 3.15.3. Assembled phage genomes were annotated manually using the Artemis [60] annotation tool. The putative open reading frames (ORFs) were predicted by using GenemarkS 4.28 [61]. Putative functions of the ORFs were analyzed by PHROGs and HHpred software [62,63]. The prediction of tRNAs was performed by using tRNAscan-SE 1.3.1 software [64]. Geneious 7.1.9 software [65] was used for multiple alignments and mapping. Comparative genomics were conducted using the Easyfig tool and VIP tree [66,67]. Genome sequence identity was calculated by VIRIDIC [68]. PhageTerm was used for the detection of phage termini. The genBank accession numbers for our nucleotide sequences were BankIt2694355 *Yersinia* OQ828305 for Phage YpYeO9 and BankIt2694380 *Yersinia* OQ828306 for Page YpEc11.

### 2.7. Influence of Temperature, pH, Osmotic Pressure, and Disinfectants on Survival of Bacteriophages

The influence of elevated temperature and variable pH on the viability of phages was studied by a standard methodology [56]. Briefly, phages (1 × 10^7^ PFU/mL) in BHI broth test tubes were incubated at various temperatures (45, 50, and 60 °C). After 10 and 30 min, the samples were collected to determine the number of infective phage particles by the soft agar overlay method [54].

SM buffer, adjusted to different pH (2, 4, 6, 8, 10, and 12), was used to study the influence of the hydrogen ion concentration on phage viability. Phages (1 × 10^7^ PFU/mL) were mixed with SM buffer with different pH and samples were collected at 15, 30, and 60 min to determine the number of infective phage particles by the soft agar overlay method [54].

The same approach was used to determine the survival rate of phages after treatment with sodium hypochlorite (5% bleach and its 1% dilution—1:5 *v*/*v* in distilled water). The number of active phage particles was determined after 1, 10, and 30 min of exposure.

The influence of ionic strength and osmotic shock on phages was studied by Anderson’s method with some modification [56]. Phages diluted in saline were added to the 3.5 M sodium chloride (NaCl) solution to obtain the concentration 1 × 10^7^ PFU/mL and were then equilibrated for 30 min at 37 °C and the number of viable phage particles was determined. Then, 0.1 mL of phage suspension in 3.5 M NaCl was quickly inoculated into a 10 mL of distilled water and samples were taken immediately for measuring of viable phage particles [54,69].

### 2.8. Subtyping of Bacterial Isolates by Phage Susceptibility Profiles

For subtyping experiments, 5 μL drops of phage suspension (1 × 10^8^ PFU/mL) were spotted on bacterial streaks on BHI agar plates. The plates were examined for lysis after 18–24 h incubation at 28 °C and 37 °C (according to the optimal growth conditions of tested bacteria). The lytic reactions were registered as described above in Section 2.4. Bacterial isolates were grouped together (phage groups) based on similar phage susceptibility profiles [70]. Individual phages were compared based on their lytic spectrum and phage susceptibility profiles within bacterial species and a phylogenetic tree was then constructed (see Section 2.10).

### 2.9. Phage Infection in Liquid Bacterial Culture (Challenge Experiments In Vitro)

The study of phage propagation in liquid microcosm was carried out in 50 mL glass flasks with 30 mL of M9 synthetic medium enriched with 0.1% yeast extract. Three flasks in each series contained (i) the target pathogen (control of bacteria); (ii) relevant phage (phage control); and (iii) a mix of phage/bacteria (challenge mixture). Experiments were conducted with phage/bacteria ratio of 100:1 and 1000:1. Samples from each tube were collected at 0, 60 min, 2 h, 4 h, and 24 h to determine the number of viable bacteria and infective phage particles by the soft agar overlay method [54].

### 2.10. Data Analysis

All measurements were carried out in triplicate for each sample. The numbers were averaged to calculate a mean value and a standard error using Windows Excel descriptive statistics program. Phylogenetic trees were constructed using FreeTree software [71] based on a distance–matrix method such as the Unweighted Pair Group Method with Arithmetic Mean (UPGMA)-based approach and phylogenetic tree visualization software (TreeView) [69,72] while VIP tree software was used to compare protein sequences [73].

## 3. Results

### 3.1. Testing of Activity of Existing Enterobacteriaceae Phages against Y. pestis EV76

Considering the possibility of cross-lytic reactivity between phages specific to different *Enterobacteriaceae* genera [35,74] we performed initial screening of existing phages from the Eliava collection, active against *E. coli*, *Shigella* spp., *Salmonella* spp., and *Y. enterocolitica* (O:3, O:5, O:6, O:8, and O:9 serotype) on the standard strain *Y. pestis* EV76. In total, 70 phages, the majority specific to *E. coli* (40 phages) and *Y. enterocolitica* (15 phages), were tested. Screenings did not reveal the development of phage plaques on the *Y. pestis* EV76 strain. Although a few lytic zones were observed initially on the bacterial lawn, no evidence of phage propagation on the target strain was later obtained. These negative results prompted us to proceed further with the search for new phages.

### 3.2. Isolation and Primery Characterization of Phages Lytic to Y. pestis EV76

Seventy-five enriched samples were obtained in order to isolate new active phages against *Y. pestis* EV76 as a host strain, as well as strains of related genera and species within *Enterobacteriaceae* (*Y. enterocolitica*, *Shigella spp*., *Salmonella spp*., and *E. coli*). For this purpose, mainly water (rivers, lakes, and city effluents) and soil samples were collected in different areas of Georgia (Tbilisi, Jandara, Sachkhere, Kazbegi, etc.). Examination of primary samples enriched with *Y. pestis* EV76 did not reveal any phage lytic activity. In contrast with these results, 54 primary phages were obtained through enrichment of environmental samples with strains of different *Enterobacteriaceae* genera. Among them, 28 primary isolates were obtained on *E. coli* strains: 6 on *S. sonnei*, 1 on *S. flexneri*, and 19 phages were obtained on *Y. enterocolitica*. These primary phage lysates underwent several series of propagation on the relevant hosts in liquid and solid media and then all new phage isolates were checked on lytic activity to the *Y. pestis* EV76 strain. Five primary phage isolates obtained on *E. coli* and one on *Y. enterocolitica* (O:9 serotype) showed lytic activity towards *Y. pestis* EV76. Finally, after a series of propagation and cloning of six primary phage isolates (some of them in mixtures) on *Y. pestis* EV76, eight phages were obtained. Among them, seven phages were originally isolated on *E. coli* and one phage on *Y. enterocolitica* (O:9 serotype). The naming of newly obtained phages lytic to *Y. pestis* EV76 was performed according to bacteriophage naming guidelines [75]. Thus, bacteriophages active against *Y. pestis* originated from *E. coli* were designated as vB_YpEc56, vB_YpEc56D, vB_YpEc57, vB_YpEc58, vB_YpEc1, vB_YpEc2, and vB_YpEc11, while phage originated from *Y. enterocolitica* were designated vB_YpYeO9.

#### 3.2.1. Phage Plaque and Virion Morphology of Bacteriophages Active against *Y. pestis*

Out of eight newly isolated phages propagated on *Y. pestis* EV76 (average titer ranging 5 × 10^9^–1 × 10^10^ PFU/mL), seven phages had formed similar plaques of a 4–6 mm diameter with a clear center and narrow turbid hallow zone. The phages YpEc2 produced plaques of the same morphology but were smaller in size (1.5–2 mm diameter).

The TEM studies demonstrated that new phages expressing lytic activity towards *Y. pestis* EV76, namely YpEc56, YpEc56D, YpEc57, YpEc58, YpEc1, YpEc2, YpEc11, and YpYeO9, have a virion morphology consistent with the podoviridae morphotype, the isometric head, and short non-contractile tail with different dimensions (Figure 1, Table 1).

#### 3.2.2. Lyses of Different Species of Enterobacteriaceae by Newly Isolated *Y. pestis* Phages

To better characterize and differentiate those eight phages active against *Y. pestis* EV76 with similar morphologies, we examined their lytic activity by a phage spot test against the set of *Enterobacteriaceae* strains from the Eliava collection of *E. coli* (21 strains), *Shigella* spp. (10), and *Salmonella* spp. (16), as well as against 45 isolates of different serotypes of *Y. enterolocitica.* In addition, the EOP was determined for randomly selected strains (Table 2, Appendix A).

The lytic activity of eight new phages active against *Y. pestis* showed a high number of susceptible hosts, lysing from 33.3% to 42.8% of *E. coli* strains. As for representatives of *Salmonella*, only four phages, YpEc57, YpEc1, YpEc56, and YpEc56D, showed activity against only *S. Enteritidis*, while the rest of the phages were inactive against all tested *Salmonella* strains. The majority of phages active against *Y. pestis* showed identical lytic profiles against *Shigella* species lysing mainly *S. sonnei* strains. Phage YpEc57 in addition to activity towards *S. sonnei* and *S. flexneri*, also lysed one isolate of *S. boydii.* The same set of phages showed more diversified lytic profiles towards *Y. enterocolitica* strains which were divided into 7 phages groups (each with 1 to 11 bacterial isolates) based on phage susceptibility profiles (Table 3).

The screening results revealed that phages YpEc2 and YpEc56D were totally inactive against the tested *Y. enterocolitica* strains. The rest of the six phages in total lysed 51.1% of strains showing different lytic profiles. In particular, phage YpEc11 covered all serotypes of *Y. enterocolitica*; considerably less active were phages YpEc57, YpEc58, and YpEc1 with each lysing strains of five phage groups. The phage YpYeO9 and phage YpEe56 showed activity toward strains of two phage groups. It should be mentioned that the strains in the majority of phage groups were lysed with several phages; only two strains in phage group seven turned out to be sensitive against phage YpEc11.

The difference in the lytic profiles of the studied eight phages in relation to the *Y. enterocolitica* strain set was also reflected in the constructed UPGMA phylogenetic tree (Appendix A).

#### 3.2.3. Lytic Activity of *Y. pestis* Bacteriophages against Strains of *Y. pestis* and *Yersinia* spp.

In order to determine the lytic activity of eight selected phages active against *Y. pestis* strains and for the selection of phages with a considerably broader host range, two different sets of *Y. pestis* strains were used for phage screenings.

Initially, *Y. pestis* phages were tested against 36 strains with different sources and dates of isolation from the collection of NCDC (Tbilisi, Georgia) (Table 4, Appendix A).

The high coverage of *Y. pestis* strains from the NCDC collection (up to 100%) was shown by the lytic activity of eight new phages active against *Y. pestis*. All strains appeared to be susceptible to each of the tested phages and moreover, the lytic reactions were of a similar grade (CL, SCL, or OL)

The host range of the tested phages was also evaluated by screening against another set of 35 strains from the IRBA collection (Bretigny sur Orge, France), including 32 *Y. pestis* strains, 2 *Y. pseudotuberculosis,* and 1 *Y. frederiksenii* (Appendix A).

Screening against the IRBA collection also showed high lytic activity (100% of *Y. pestis* tested strains) of all eight phages active against *Y. pestis (*Table 5 and Appendix A). On both sets of strains, despite different geographic origins and sample types, phages revealed a high intensity of lytic reaction (CL type). All phages but one (YpEc56D) were active against the strain of one out of the two *Y. pseudotuberculosis* tested strains, while the *Y. frederiksenii* strain appeared to be resistant to all screened phages. However, considering the low number of tested strains, such findings cannot be generalized for the entire population of these species.

The lytic activity of the same eight *Y. pestis* phages was also evaluated by EOP for randomly selected *Yersinia* spp. strains from both NCDC and IRBA collections. It was shown that the CL type of the lytic reaction corresponded to a high EOP (0.5–1.0), OL, and SCl-to a medium EOP (0.1–0.5); and IP to a low EOP (0.001–0.1) (Appendix A).

After three rounds of screenings, two phages, YpYeO9 and YpEc11, were selected for the next step, aiming for detailed studies of phages active against *Y. pestis* expressing therapeutic potential. The selection was performed according to the following characteristics: (i) different sources (samples) of isolation and primary host strains (species) (YpYeO9 was isolated on *Y. enterocolitica* serotype O:9 and YpEc11 was isolated on *E. coli* C) and (ii) different profiles of lytic activity towards the *Y. enterocolitica* strains (phage YpEc11 showed a broad lytic spectrum (lysed strains of all phage groups) while phage YpYeO9 showed activity only against three phage groups).

### 3.3. Detailed Characterization of Selected Bacteriophages Active against Y. pestis

#### 3.3.1. Whole Genome Sequencing of Phage Active to *Y. pestis* and Analysis

Phage YpYeO9 contains dsDNA and is 38,761 bp in length, CG content is 48.74%. A total of 33,708 bp were identified as the gene coding region and in total, 41 ORFs were predicted. Short (148 bp) direct terminal repeats were identified. Among 41 ORFs, 34 were functionally annotated. Genes with predicted functions were grouped as follows: head and packaging module (9 predicted genes); tail morphogenesis module (4 predicted genes); host cell lysis module (3 predicted genes) encompassing DNA and RNA metabolism (11 predicted genes), 1 CDS was annotated as a head to tail connection, and 6 CDSs were identified as auxiliary metabolic and host takeover genes (Figure 2). The phage genome does not contain lysogeny control genes and no tRNAs were identified. Phage YpYeO9 was identified as a T7-like phage and member of *Teseptimavirus* (Appendix A).

A blast similarity search revealed that YpYeO9 is highly similar to the phages YpP-R and *Enterobacter* T7 (NC_001604.1). Studies of average nucleotide identity show that ANI are 87% (YpP-R) and 88.5% (T7) which indicates that YpYeO9 belongs to a new species (Figure 3).

Phage YpEc11 contains dsDNA, is 39,896 bp in length, and the CG content is 45.95%. A total of 35,604 bp were identified as gene coding region and a total of 44 ORFs were predicted. Among 44 ORFs, 30 were functionally annotated. Genes with predicted functions were grouped as follows: head and packaging module (9 predicted genes); tail morphogenesis module (3 predicted genes); host cell lysis module (2 predicted genes) encompassing DNA and RNA metabolism (11 predicted genes), 1 CDS was annotated as a head to tail connection, and 4 CDSs were identified as auxiliary metabolic and host takeover genes (Figure 4). The phage genome does not contain lysogeny control genes and no tRNAs were identified. Phage YpEc11 is identified as a member of *Helsettvirus* (Appendix A).

Blast similarity search revealed that YpEc11 is highly similar to the phage fPS-59. A study of the average nucleotide identity shows that ANI for those phages is 82.7% which indicates that YpEc11 belongs to a new species (Figure 3).

A study of comparative genomics showed that Phages YpYeO9 and YpEc11 have a similar genome organization as well as the following phages: phage Berlin, Yepe2, fPS-59, phiA1122, T7, and T3 but are completely different to L-413C. The results were visualized by easy fig software (Figure 5).

Comparative genomics results are fully compliant with the results of the viral proteomic tree, constructed with VIP tree software (Figure 6).

#### 3.3.2. Lysis Stability of Selected Bacteriophages Active against *Y. pestis*

The stability of phages active against *Y. pestis* in liquid culture, namely YpYeO9 and YpEc11, was determined by the Appelmans method [58] using phage suspensions with the initial titer of 2 × 10^9^ PFU/mL. Both phages showed high lysis stability (Table 6). At all phage–bacteria ratios, the clear lysis was registered during 24 h of incubation. After 48 h, both phages maintained the stabile lyses at all MOI’s and this was maintained for up to one week in the case of phage YpEc11. The visible bacterial growth was detected in reaction tubes with YpYeO9 phage–bacteria ratio 0.0001 but was still less than in control tubes. The corresponding phage titer by Appelmans comprised 10^−8^ in 24 h and 10^−7^–10^−8^ in 48 h. The prolonged lysis stability maintained for up to one week is characteristic of virulent phages.

#### 3.3.3. Viability of Phages in Different Solutions

The study of the viability and stability of phages is highly important to assess their capabilities as infection control agents. The experimental investigations showed that titers of phages YpYeO9 and YpEc11 in BHI broth, SM buffer, and PSB solution at 4 °C were stable even after a 6-month period. The initial phage titer was decreased by 2–3 logs only after a 1-year period (Figure 7).

#### 3.3.4. Frequency of Formation of Phage Resistant Mutants

Determination of the frequency of formation of phage-resistant mutants is an important characteristic of phage activity that is usually linked with parameters of lysis stability in liquid culture. The experiments involving phage YpYeO9 and YpEc11 were performed on a solid medium according to the methodology of Chanishvili and Kapanadze (1967). Both phages demonstrated a low frequency (1–2 × 10^−6^) of formation of phage-resistant mutants. The obtained results in addition to the phage lysis stability data (Section 3.3.2) indicate the virulent nature of studied phage.

#### 3.3.5. The Phage One-Step Growth Cycle

The steps of phage–host interactions (adsorption, latent period, and burst size) for phages YpEc11 and YpYeO9 were studied. The parameters of one step growth cycle (OSGC) for both phages were determined on *Y. pestis* EV76 and on the primary hosts: *E. coli* C (for the phage YpEc11 propagated on *E. coli* C and designated as YpEc11*E) and *Y. enterocolitica* O:9 (for the phage YpYeO9 propagated on *Y. eneterocolitica* O:9 and designated as YpYeO9*Ye).

The experiments conducted on *Y. pestis* EV76 showed that for both phages, namely YpEc11 and YpYeO9, the maximum adsorption time was 12 min with adsorption efficacies of 74.5% and 68.1%, respectively. The latent period for YpEc11 was 90 min and 60 min for phage YpYeO9. The burst size for the phage YpYeO9 was shown to be 85 PFU per infected cell, which was higher in comparison with the burst size of phage YpEc11: 52 PFU per infected cell (Figure 8).

The adsorption parameters for phage YpEc11*E propagated on primary host *E. coli* C and for the same phage propagated on *Y. pestis* EV76 strain (phage YpEc11) were shown to be very similar regarding the adsorption time (12 min) and efficacy (70.3–74.5% for YpEc11*E and YpEc11, respectively). The latent period for the YpEc11*E phage was shorter: 60 min and the burst size was bigger (72 PFU per infected cell) than for phages propagated on *Y. pestis* EV76 cells.

The adsorption parameters for phage YpYeO9*Ye propagated on primary host *Y. enterocolitica* O:9 serotype differed from the parameters for the same phage propagated on *Y. pestis* EV76 strain (phage YpYeO9). In particular, the adsorption time for phage YpYeO9*Ye was 8 min, with adsorption efficacy 87%. The latent period for YpYeO9*Ye phage was shorter: 40 min and the burst size was bigger (92 PFU per infected cell) than for phages propagated on *Y. pestis* EV76 cells.

### 3.4. Stability and Infectivity of Phages Active against Y. pestis in Different Environmental Conditions

#### 3.4.1. Influence of Acidic and Alkaline Environment on Survival of Phage Active against *Y. pestis*

The sensitivity of selected phages active against *Y. pestis* to various pH was studied for 30 and 60 min exposure times (Figure 9).

Both phages, YpYeO9 and YpEc11, showed their enhanced susceptibility to strong alkaline conditions (pH12) rather than to strong acidic conditions (pH2). Namely, at pH 2 the titer of both phages was reduced by 2–3 log after 30-min and by 3–4 log after 60 min exposure, respectively. In the strong alkaline conditions (at pH 12), the titer of phage YpYeO9 decreased by up to 6 log after 30 and 60 min exposure, while in the case of YpEc11 the viable phage count was reduced by 6.5 log in 30 min and was not detectable in 60 min.

#### 3.4.2. Influence of Temperature on Survival of Phage Active against *Y. pestis*

The experiments on the thermal inactivation of phages active against *Y. pestis* were conducted by counting viable phages after 10 and 30 min exposure to the increasing temperature in the range from 28 °C to 60 °C (Figure 10).

The optimal growth temperature for both phages YpYecO9 and YpEc11 was found to be 28 °C. The same phage titer was maintained at 37 °C and 45 °C for 10 min, but prolonged exposure time (30 min) led to 1 and 1.5 log reductions in phage counts, respectively. Both 10 and 30 min exposures at 50 °C resulted in an equal decrease (by 1.5 log) in viability for the phage YpYeO9. As for phage YpEc11, the phage titer was decreased by 1.5 log in 10 min and by 3 log in 30 min. Further increases in temperature (60 °C) during 10 min led to a reduction of viable counts of YpYeO9 and YpEc11 phages by 4–5.5 log, respectively, while a 30 min exposure decreased the number of viable phage particles below the detectable level.

#### 3.4.3. Influence of High Ionic Strength on Survival of Phage Active against *Y. pestis*

The selected phage were tested for resistance to high ionic strength. It was shown that the titers of YpYeO9 and YpEc11 phages were decreased by 2.6 logs and 3.0 logs, respectively, after 30 min exposure to the 3.5 M NaCl solution (Table 7).

In the next stage, after the addition of distilled water to the reaction tubes with a 3.5 M NaCl solution containing phages YpYeO9 and YpEc11, both phages maintained their titer. Despite the rapid and sharp decrease in osmotic pressure, the viability of the studied phages remained unaffected.

#### 3.4.4. Influence of Disinfectant on the Survival of Phages Active against *Y. pestis*

The testing of selected phages YpYeO9 and YpEc11 for sensitivity to commonly used disinfectant, namely sodium hypochlorite (NaOCl), showed a reduction of the phage infectivity by 3.5 log in the case of YpYeO9 and by 5 log for the phage YpEc11 after 1 min of exposure to the 1% solution of NaOCl (Figure 11).

The 10 min exposure to the same disinfectant did not reduce the viability of YpYeO9 and YpEc11 phages more: their titer remained practically the same while treatment of both phages during 30 min totally eliminated the viable phage particles. As for treatment with a 5% solution of NaOCl, the total inactivation of both phages was observed almost immediately, within 1 min (results not included in Figure 11).

#### 3.4.5. Antibacterial Efficacy of YpYeO9 and YpEc11 Phage in Liquid Culture

Experimental infections of *Y. pestis* EV 76 in liquid culture with phages YpYeO9 and YpEc11 and a mixture of these phages at different phage/bacteria (P/B) ratios were carried out. In the case of the P/B ratio of 100:1 (Figure 12a), a gradual reduction in bacterial counts was observed during the first 4 h (2.5 log reduction for phage YpE11, 1 log for phage YpYeO9, and 3.6 log for the YpYeO9/YpEc11 mixture) although a considerably sharp decline of viable bacterial cells was demonstrated initially for the phage mixture. After 24 h of exposure, similarly as for individual phages and their mixture, all bacteria were found to be destroyed (the number of viable particles was reduced by 5 log and was below the detectable level). It should be noticed that the number of viable bacteria in the *Y. pestis* EV76 control was reduced by >1 log after 4 h and maintained at this level for 24 h.

A much higher antibacterial efficacy was shown in the case of a P/B ratio of 1000:1 (Figure 12b). Similarly, for both individual phage and their mixture up to 2.5 log, a reduction in comparison with bacterial control was shown in 2 h. After 4 h, only a 3 log reduction in bacterial counts was registered for individual phages, while for the phage mixture no viable bacteria were detected. For individual phages, the total disappearance of bacteria was observed after 24 h.

## 4. Discussion

The plague still causes outbreaks worldwide, mainly in Africa, but also in Asia and South America. The majority of human plague cases reported in the last two decades occurred in small towns and villages or agricultural areas [76]. The treatment and prevention of the plague is still a challenge.

The goal of the presented study was to isolate and characterize new phages active against *Y. pestis* in order to determine their potential for practical applications such as the detection/identification and subtyping of *Y. pestis* strains, infection control, and therapeutics.

The initial steps for phage isolation from water and soil samples on the host strain *Y. pestis* EV76 were not successful despite intensive sampling and isolation attempts. This can be explained firstly because the vaccine strain *Y. pestis* EV76 was the only target strain available for the work in the BSL2 labs for isolation and propagation of *Y. pestis* active phages. Secondly, the reason could be the low circulation of *Y. pestis* bacterium in the environment of Georgia during the last two decades, which usually is reflected in a low abundance of corresponding specific phages [3].

Considering these circumstances, we proceeded with an indirect way of obtaining active phages for *Y. pestis* by targeting related *Enterobacteriaceae* species such as *E. coli*, *Shigella* spp., *Salmonella* spp., and *Y. enterocolitica*. We started with screening existing phages active against the above-mentioned species towards *Y. pestis* EV76. In total, 70 phages with primary specificity to *Enterobacteriaceae* species were tested but none of them showed lytic activity on *Y. pestis* EV76.

Another approach intended to isolate new phages from environmental samples by using the newly isolated as well as standard strains of *E. coli*, *Shigella* spp., *Salmonella* spp., and *Y. enterocolitica* as host strains in enrichment experiments. As a result, 54 primary phage isolates were obtained and the lytic activity towards *Y. pestis* EV76 was expressed by 8 phages, 7 of which were primarily active against *E. coli* and 1 to *Y. enterocolitica* O:9. These eight phages, after propagation and purification on the *Y. pestis* EV76 strain, were studied for a number of phenotypic characteristics as well as for genetic properties.

The primary comparative characterization of new phages active against *Y. pestis* was conducted based on phage plaque and nucleocapsid morphology as well as on lytic activity against a set of strains of different *Enterobacteriaceae* genera. The obtained results showed quite a low diversity among studied *Y. pestis* phages. All 8 phages have similar plaque morphologies on the host strain *Y. pestis* EV76 with a clear center and narrow hollow zone, but slightly different in size. The virion morphology of each of the eight studied phages was consistent with the podoviridae morphotype of tailed bacteriophages [77]. Phages active against *Y. pestis* with the same morphology were described in different scientific papers [37,53]. The phage host range is a very important characteristic used for the selection of candidate therapeutic bacteriophages. Eight isolated phages active against *Y. pestis* were tested for antibacterial activity in vitro against two sets of *Y. pestis* strains: from collections IRBA (Bretigny sur Orge, France) and NCDC (Tbilisi, Georgia), as well as a mixed set of strains of *E. coli*, *Shigella* spp., and *Salmonella* spp. and a set of *Y. enterocolitica* strains with different serotypes (collection of the Eliava Institute, Georgia). An extremely broad host range and similarly practically identical lytic profile for all 8 *Y. pestis* phages was shown towards *Y. pestis* isolates from strain collections of NCDC and IRBA. As for the species specificity of our phages, all 8 phages did not show activity against *Y. frederiksenii* (1 IRBA strain) while 1 strain of *Y. pseudotuberculosis* (out of 2 IRBA strains) appeared to be sensitive towards the majority of phages. As a result of the limited number of strains of the above-mentioned species, we cannot provide a statistically reliable analysis of the obtained data. According to other studies [78], some *Y. pestis* phages can be strictly specific to *Y. pestis*, while the host range of other phages included also *Escherichia coli* and *Yersinia pseudotuberculosis*.

The determination of the phage host range is important not only for therapeutic purposes but also for subtyping of bacterial isolates. Phage typing was widely used to identify and distinguish different strains within a given species when isolated from various sources (clinical sample, food, water, and environmental) or geographical locations [32,33,34].

A very broad spectrum of tested phage revealed in the screenings on *Y. pestis* strains indicated the low potential of application of these phages for grouping of *Y. pestis* strains by phage susceptibility profiles and for further development of a. phage-based subtyping set for *Y. pestis*. A different and more promising situation was shown in the case of *Y. enterocolitica* isolates of different serotypes against which all eight phages active against *Y. pestis* showed diversified action. This allowed dividing these strains into eight phage groups.

At the same time, based on the extremely broad lytic spectrum of the new phages active against *Y. pestis*, we can propose their high potential for application as an effective and ecologically safe means for infection control, particularly for the cleaning/decontamination procedures to be conducted in different systems. Phages active against *Y. pestis* with a high antibacterial potential have also been described by other authors [79].

The studied eight new phages did not reveal a diversified action in screenings against a set of *E. coli*, *Shigella* spp., and *Salmonella* spp. strains. Namely, all tested phages appeared to be active towards *E. coli*, *S. sonnei,* and *S. flexneri* strains, but only three out of eight phages—YpEc57, YpEc56, and YpEc56D—demonstrated lytic activity towards *Salmonella*, particularly *S. enteritidis* and *S. typhimurium*. A similar spectrum of lytic activity for *Y. pestis* phages was previously described by other investigators [78,80,81]. Interestingly, the host range for some *Y. pestis* phages was restricted to *Y. pestis* strains, while some others included different members of *Enterobacteriaceae* such as *E. coli*, *Salmonella* spp., *Shigella* spp., *Y. enterocolitica,* and *Y. pseudotuberculosis.*

Further experimental studies aimed at the detailed phenotypic and genetic characterization of phages, including full genome sequencing, were performed on two phages: YpYeO9 and YpEc11. These phages were selected based on different primary bacterial hosts used for their isolation (*E. coli* and *Y. enterocolitica*) and also on different phage lytic profiles demonstrated by these phages on the set of *Y. enterocolitica* strains.

A study of biological properties of the new phages active against *Y. pestis*, including the initial steps of the host–phage interaction, frequency of formation of phage-resistant mutants, and durability under different environmental conditions (pH, temperature, ionic strength, disinfectant, and viability in different solutions) are important for the assessment of their potential for practical application as antibacterial agents to be used for control of infections caused primarily by *Y. pestis*. Reported lytic *Yersinia* phages are mostly *Podoviruses*, as our new phages, although *Myoviruses* were also described, such as PST, ΦJA1, PY100, JC221, and fD1 and a novel type of dwarf *Myovirus* fEV-1 [22,50,52,78]. Until now, the majority of existing *Y. pestis* phages are routinely used for the diagnosis of plague. As for phage application as therapeutic agents for *Y. pestis* infection, a number of experimental animal studies were conducted. Vagima et al. (2022) used a mouse model for pneumonic plaque to assess the phage therapy potential using known phages ΦA1122 and PST [26]. Phage application significantly delayed mortality and limited bacterial proliferation in the lungs but could not prevent bacteremia. To compensate for the certain insufficiency in treatment, the combination of these phages with the antibiotic ceftriaxone was used which led to the survival of all infected animals, thus demonstrating a synergistic protective effect. For use of phages for the treatment of plague or any other serious bacterial infection, besides some important phenotypic features (host range, lysis stability, etc.) they need to be proven to be lytic and to be stable under different physical–chemical conditions.

For therapeutic applications, the knowledge of the genomic characteristics of candidate phages is especially important to avoid carriage of certain genes encoding virulence factors, integrases, etc. Genome sequencing showed that phage YpYeO9 contains dsDNA of 38,761 bp in length with GC content 48.74% and the phage YpEc11 also contains dsDNA of a similar size (39,896 bp in length) and lower GC content of 45.95%. Genome sequencing demonstrated that genomes of YpYeO9 and YpEc11 phages do not contain lysogeny control genes and no genes encoding tRNAs, which indicates the lytic-only nature of both new phages lysing *Y. pestis* strains. Phage YpYeO9 was identified as a T7-like phage, member of *Teseptimavirus* genus of the order *Caudovirales*, in the family *Autographiviridae*, and in the subfamily *Studiervirinae.* The phage YpEc11 was attributed to the *Helsettvirus* genus of the same order and family. Both phages based on the Blast similarity search were highly similar to phages YpP-R, Enterobacter T7 (NC_001604.1), and fPS-59, but nucleotide identity showed that phages YpYeO9 and YpEc11 belong to a new viral species. A study of comparative genomics showed that phages YpYeO9 and YpEc11 have genome organizations similar to phages Berlin, Yepe2, phiA1122, T7, and T3, while they are completely different from L-413C phage. This study also enabled the determination as to whether our phages contained any genes that are considered “undesirable” for phage preparations to be used for food safety, surface decontamination, and human clinical applications [37]. None of our two fully sequenced phage contain genes encoding integrases and also genes responsible for bacterial virulence factors.

The lytic nature of YpEc11 and YpYeO9 phages is in agreement with their phenotypic characteristics, such as the high lysis stability (up to 1 week) in liquid culture and low frequency of formation of phage-resistant mutants (1–2 × 10^−6^). The one-step growth cycle parameters for phages YpYeO9 and YpEc11 showed considerably short adsorption times (12 min) and moderate to high adsorption efficacies (68.15–74.5%), followed by a 60–90 min latent period and sufficient burst size (85 PFU–52 PFU, respectively). These results are within the range of infection kinetics data, somewhere in the middle, obtained for other phages lytic to the *Enterobacteriaceae* family. For example, the burst size of *Podovoridae* phages of *Enterobacteriacerae* was reported to be as low as 12 PFU [82] and as high as 187 PFU/cell [83,84]. Phages with different morphotypes, for example T4-like phages active against *Y. pseudotuberculosis,* have a latent period of 50–55 min and burst size 44–65 PFU [85].

Reviewing other phenotypic properties, which are considered important for practical applications [86], we should mention the high stability of YpEc11 and YpYeO9 phages in BHI broth, SM buffer, and PSB during a 6-month period, their viability under acidic and moderately alkaline conditions, and their resistance to drastic changes in osmotic pressure. In addition, moderate thermo-tolerance (up to 60 °C) of tested phages was shown as was their considerable durability against acidic conditions (especially phage YpeYeO9), rather than strong alkaline conditions.

We studied the sensitivity of *Y. pestis* phages to commonly used disinfectants in clinical and laboratory practices such as bleach (sodium hypochlorite) which has strong sporocidal and virucidal activities. The high susceptibility of YpEc11 and YpYeO9 phages to 1% and 5% concentrations of NaOCl was shown to be comparable with the inactivation rate of Picornaviruses, non-enveloped small size icosahedral viruses [87,88]. Interestingly, the host organism, *Y. pestis*, known as a non-spore-forming bacterium, is highly susceptible to similar concentrations of sodium hypochlorite [89,90].

To assess the potential of YpYeO9 and YpEc11 phages and their mixture for efficient biocontrol (rapid elimination) of *Y. pestis* infection in a water environment, the experimental studies were performed in a liquid microcosm with *Y. pestis* EV76. We conducted experiments in the semisynthetic M9 medium at 28 °C in the conditions that can ensure the propagation of *Y. pestis.* By observation of other investigators [91,92], in a nutrient restricted environment such as tap water and at low temperature *Y. pestis* can enter a viable but non-culturable (VBNC) state, while its supply with sufficient nutrients supports not only persistence but also propagation of *Y. pestis* in the water samples even at 4 °C. In our experiments, we used an inoculum of *Y. pestis* of 10^5^ CFU/mL (final concentration in the microcosm) that can be comparable with natural microbial contamination levels (10^3^–10^6^ CFU/mL) of surface waters [93,94] and also is in the same range used in the above-mentioned studies on *Y. pestis* survival and propagation in water environments [91,92]. For the final outcome, the killing effect of phages the phage/bacteria (P/B) ratio as well as the combined activity of two phages was found to be important. In the case of a P/B ratio of 100:1, more gradual reductions in bacterial counts in the first 4 h were observed for individual phages, but by 24 h the total clearance of bacteria was shown for YpYeO9 and YpEc11 phages and their mixture. Much higher antibacterial efficacy was registered in experiments with the P/B ratio 1000:1 although there was a difference in time required for a total reduction in the number of bacterial counts. A faster decrease in viable bacterial counts was achieved by a mixture of phage (4 h) compared to individual phage (24 h). Although in the conditions of high MOIs we cannot exclude the possibility of lysis without (LO), considering the lysis development in 4 h (for phage mixture) and 24 h (for individual phages) it would be challenging to classify this as rapid premature lysis that is one of the main attributes of the LO [95]. It should be mentioned that efficient phage-based removal of enteric bacteria in liquid systems and from solid surfaces at high MOI was demonstrated by Turki et al., 2012, and Abuladze et al., 2008 [30,96]. The MOI of 100:1 and 1000:1 was also shown to be effective in our previous studies on the application of phages in water microcosms and glass surfaces contaminated with *S. aureus* and *P. aerugionosa* [97].

The results presented and discussed above describe the phenotypic characteristics as well as genomic parameters of two new phages with podoviral morphotypes, namely YpYeO9 and YpEc11, which were initially isolated on *E. coli* and *Y. enterocolitica* and subsequently propagated on *Y. pestis* EV76. Based on genome analysis, these two phages were shown to be a new species within the family *Autographiviridae.* The YpYeO9 and YpEc11 phages demonstrated a broad lytic spectrum and wide range of *Y. pestis* strains and still showed activity to strains of some other *Enterobacteriaceae* species. These properties in conjunction with the peculiarities of phage–host interactions, viability under different physical–chemical factors, and strong antibacterial activity in liquid culture indicate the potential of YpYeO9 and YpEc11 phages for their practical use as antibacterial agents and means for reduction or elimination of highly pathogenic bacteria such as *Y. pestis* from contaminated environments.

## Figures and Tables

**Figure 1 viruses-15-01484-f001:**
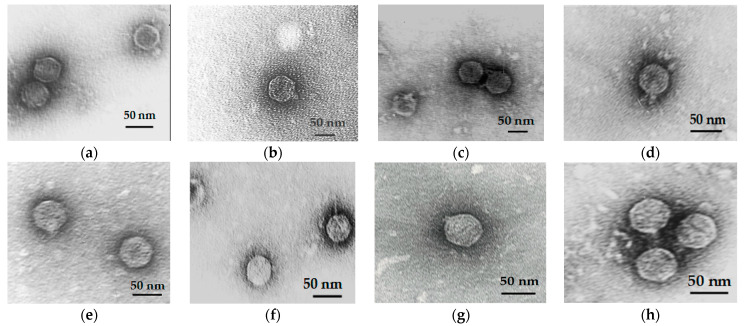
Virion morphology of phage active against *Y. pestis*: (**a**) YpEc1; (**b**) YpYeO9; (**c**) YpEc56D; (**d**) YpEc58; (**e**) YpEc57; (**f**) YpEc2; (**g**) YpEc11; and (**h**) YpEc56. TEM JEM 100SX (JEOL), instrumental magnification × 40,000.

**Figure 2 viruses-15-01484-f002:**
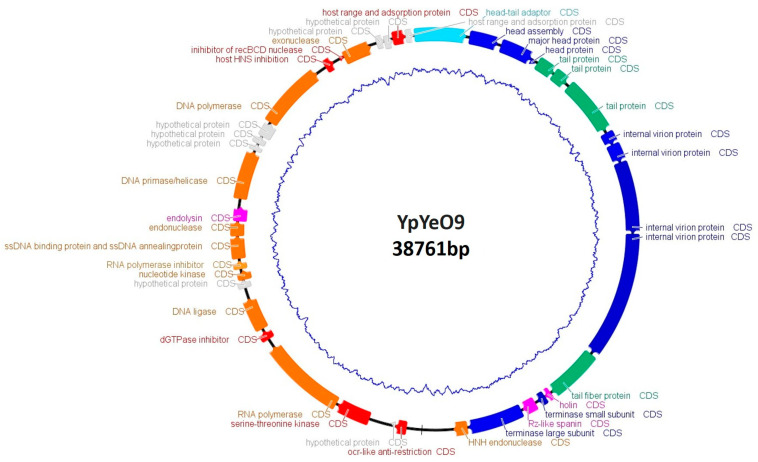
Genome map of phage YpYeO9. Gene functions are color-coded (orange: DNA and RNA metabolism; green: tail; blue: DNA packaging and head; light blue: head to tail; fuchsia: lysis; red: auxiliary metabolic and host takeover genes; gray: hypothetical protein).

**Figure 3 viruses-15-01484-f003:**
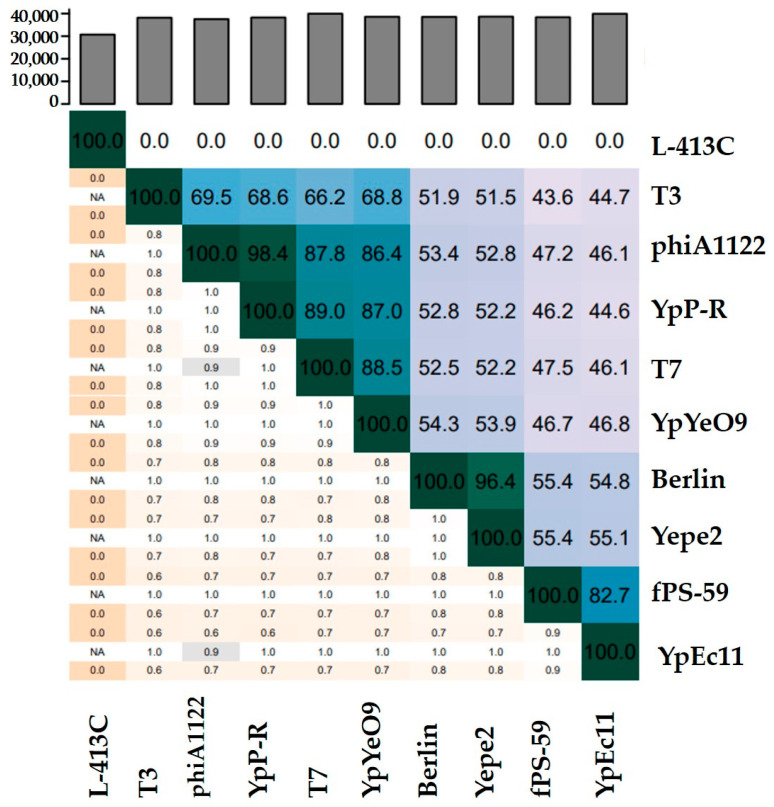
Intergenomic similarities heat map constructed by VIRIDIC showing the average nucleotide identity between the following phages: YpYeO9, YpEc11 phage Berlin (NC_008694.1), Yepe2 (NC_011038.1), phiA1122 (AY247822.1), T7 (NC_001604.1), T3 (NC_047864.1), fPS-59 (NC_047935.1), and YpP-R (JQ965701.1). Indeed, they are completely different to L-413C (NC_004745.1).

**Figure 4 viruses-15-01484-f004:**
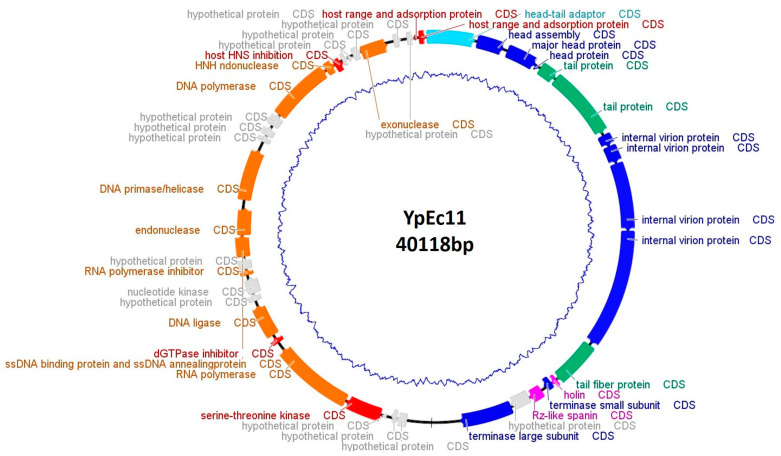
Genome map of phage YpEc11. Gene functions are color-coded (orange: DNA and RNA metabolism; green: tail; blue: DNA packaging and head; light blue: head to tail; fuchsia: lysis; red: auxiliary metabolic and host takeover genes; gray: hypothetical protein).

**Figure 5 viruses-15-01484-f005:**
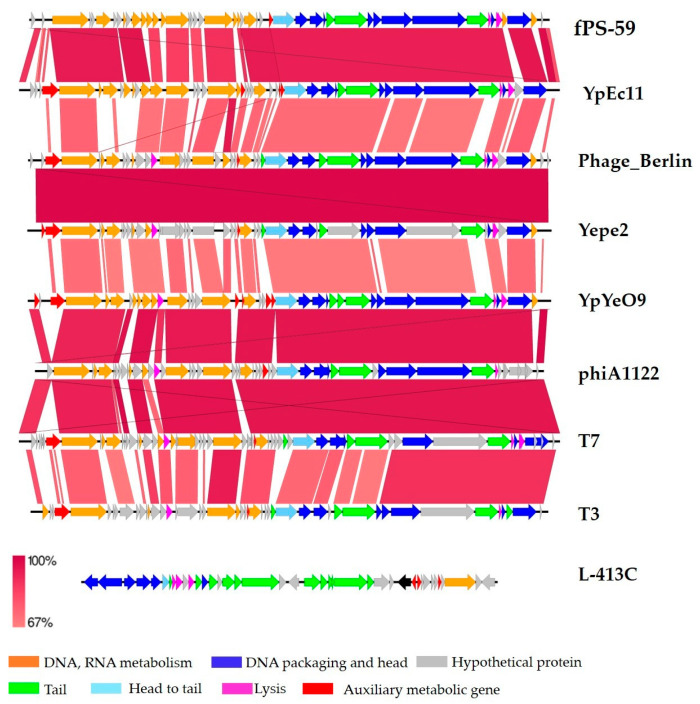
Annotation and comparison of the phages’ genomes. Gene functions are color-coded (orange: DNA and RNA metabolism; green: tail; blue: DNA packaging and head; light blue: head to tail; fuchsia: lysis; red: auxiliary metabolic and host takeover genes; gray: hypothetical protein).

**Figure 6 viruses-15-01484-f006:**
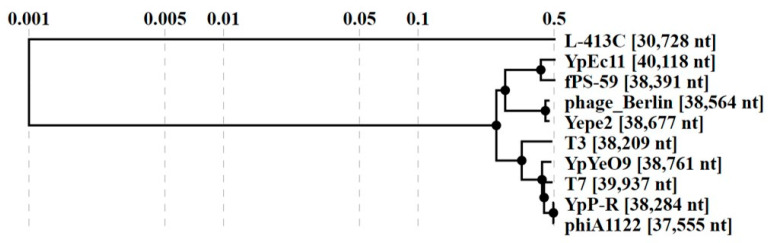
The viral proteomic tree constructed with the VIP tree.

**Figure 7 viruses-15-01484-f007:**
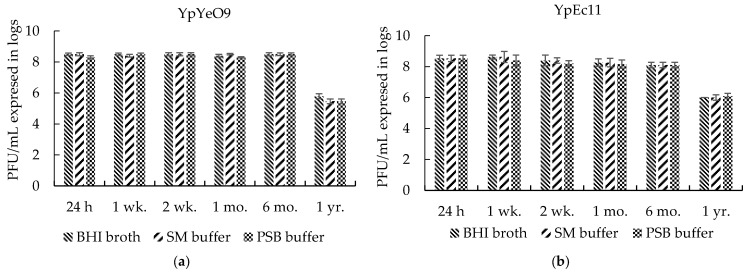
The viability of phage active against *Y. pestis* in BHI broth, SM buffer, and PSB solution: (**a**) phage YpYeO9 and (**b**) phage YpEc11. The results are the averages of three parallel experiments with geometric SD shown as the vertical lines.

**Figure 8 viruses-15-01484-f008:**
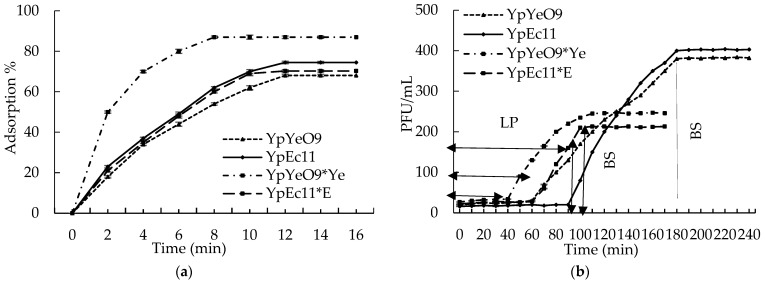
Phage–host interaction: (**a**) adsorption curve of YpEc11, YpYeO9, YpEc11*E, and YpYeO9*Ye phage and (**b**) single-step growth curve of YpEc11 and YpYeO9 phage. The results are the mean values of three independent tests. Standard deviations (SD) are indicated.

**Figure 9 viruses-15-01484-f009:**
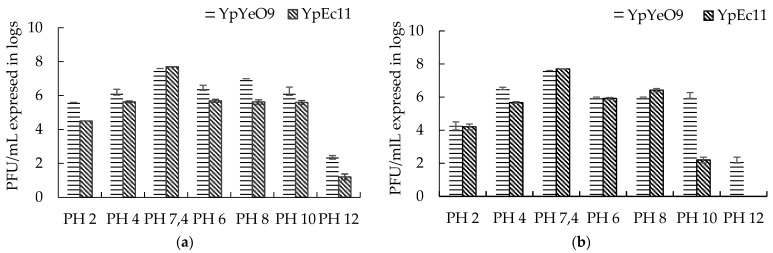
Survival of phage active against *Y. pestis* in acidic and alkaline environments during different exposure times: (**a**) 30 min exposure and (**b**) 60 min exposure. The results are the averages of three parallel experiments with geometric SD shown as the vertical lines.

**Figure 10 viruses-15-01484-f010:**
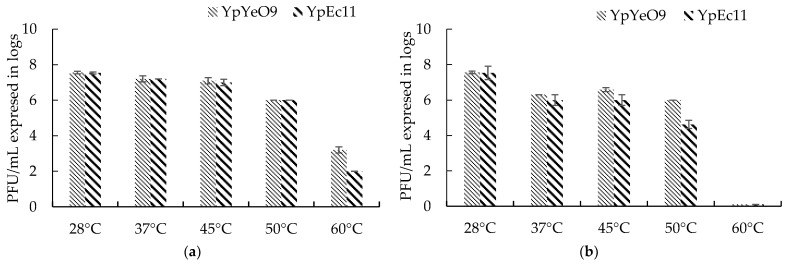
The survival of phage active against *Y. pestis* at different temperatures at (**a**) 10 min exposure and (**b**) 30 min exposure. The results are the averages of three parallel experiments with geometric SD shown as the vertical lines.

**Figure 11 viruses-15-01484-f011:**
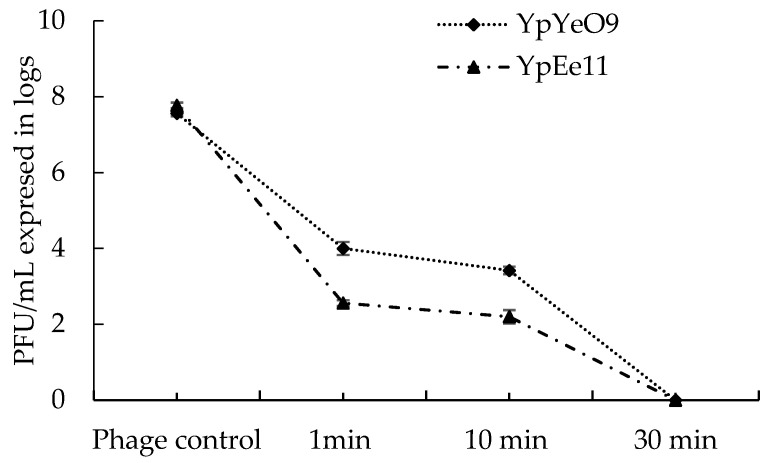
The survival of phage active against *Y. pestis* in 1% sodium hypochlorite solution. The results are the averages of three parallel experiments with geometric SD shown as the vertical lines.

**Figure 12 viruses-15-01484-f012:**
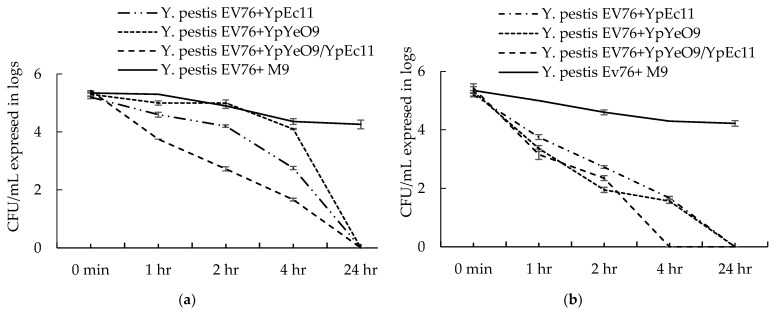
Antibacterial efficacy of phage active against *Y. pestis* in liquid microcosm: (**a**) phage/bacteria 100:1 and (**b**) phage/bacteria 1000:1. The results are the averages of three parallel experiments with geometric SD shown as the vertical lines.

**Table 1 viruses-15-01484-t001:** Virion size and morphology of eight phage active against *Y. pestis*.

Phage Active against *Y. pestis*	Morphotype	Phage Virion Size	
		Head nm	Tail nm
YpEc57	podovirus	60 ± 2 × 60 ± 2	8 ± 1 × 4 ± 1
YpEc58	podovirus	50 ± 1 × 50 ± 1	18 ± 1 × 8 ± 1
YpEc2	podovirus	48 ± 1 × 48 ± 1	8 ± 1 × 4 ± 1
YpEc11	podovirus	60 ± 2 × 60 ± 2	12 ± 1 × 5 ± 1
YpYeO9	podovirus	60 ± 2 × 60 ± 2	16 ± 1 × 10 ± 1
YpEc1	podovirus	48 ± 1 × 48 ± 1	8 ± 1 × 4 ± 1
YpEc56	podovirus	55 ± 1 × 53 ± 1	18 ± 2 × 10 ± 2
YpEc56D	podovirus	59 ± 2 × 59 ± 2	18 ± 2 × 10 ± 2

**Table 2 viruses-15-01484-t002:** Lytic activity of phages active against *Y. pestis* against different species of *Enterobacteriaceae*.

Bacterial Strains	Total Number of Isolates	Number of Strains Lysed by Phages Active against *Y. pestis*							
		YpEc57	YpEc58	YpEc2	YpEc11	YpYeO9	YpEc1	YpEc56	YpEc56D
*E. coli*	21	8/21	8/21	7/21	7/21	7/21	9/21	9/21	9/21
*S. Typhimurium*	6	0/6	0/6	0/6	0/6	0/6	0/6	0/6	0/6
*S. Agona*	4	0/4	0/4	0/4	0/4	0/4	0/4	0/4	0/4
*S. Enteritidis*	4	1/4	0/4	0/4	0/4	0/4	1/4	1/4	0/4
*S. Paratyphi A*	1	0/1	0/1	0/1	0/1	0/1	0/1	0/1	0/1
*S. Oranienburg*	1	0/1	0/1	0/1	0/1	0/1	0/1	0/1	0/1
*S. sonnei*	4	4/4	3/4	3/4	3/4	3/4	3/4	3/4	3/4
*S. flexneri*	5	1/5	1/5	1/5	1/5	1/5	1/5	1/5	1/5
*S. boydii*	1	1/1	0/1	0/1	0/1	0/1	0/1	0/1	0/1
*Y. enterocolitica* serotype O:9	10	10/10	10/10	0/10	10/10	10/10	10/10	2/10	0/10
*Y. enterocolitica* serotype O:8	6	1/6	0/6	0/6	1/6	0/6	1/6	0/6	0/6
*Y. enterocolitica* serotype O:3	9	4/9	4/9	0/9	5/9	1/9	4/9	2/9	0/9
*Y. enterocolitica* serotype O:6	11	2/11	1/11	0/11	3/11	1/11	2/11	0/11	0/11
*Y. enterocolitica* serotype O:5	9	3/9	3/9	0/9	4/9	2/9	3/9	0/9	0/9

**Table 3 viruses-15-01484-t003:** Grouping of *Y. enterocolitica* strains of different serotypes by susceptibility profiles to phages active against *Y. pestis*.

Phage Group	*Y. enterocolitica* Strains	Susceptibility to *Y. pestis* Phage							
		YpEc57	YpEc58	YpEc2	YpEc11	YpYeO9	YpEc1	YpEc56	YpEc56D
1	O:9/8377, O:9/3289, O:3/7077								
2	O:9/806, O:9/38, O:9/16, O:9/208, O:9/162, O:9/718, O:9/426, O:9/3228, O:5/171, O:5/176, O:6/183								
3	O:3/7577								
4	O:3/121, O:3/19								
5	O:8/675, O:6/268, O:5/270								
6	O:5/104								
7	O:3/176, O:6/261								

A grey box indicates a positive result of phage lytic activity: (CL, SCL, and OL) and an uncolored box indicates a negative result. The total number of *Y. enterocolitica* strains was 45, out of which 22 strains appeared to be non-susceptible to any of the tested phages.

**Table 4 viruses-15-01484-t004:** Lytic activity of phages active against *Y. pestis* against set one of *Y. pestis* strains (Collection of NCDC Tbilisi, Georgia).

Bacterial Strain	Source of Isolation	Total Number of Strains	Susceptible to Phage Active against *Y. pestis*							
			YpEc57	YpEc58	YpEc2	YpEc11	YpYeO9	YpEc1	YpEc56	YpEc56D
*Y. pestis*	*M. arvalis*	9	9/9	9/9	9/9	9/9	9/9	9/9	9/9	9/9
	*C. caspius*	7	7/7	7/7	7/7	7/7	7/7	7/7	7/7	7/7
	*C. teres*	20	20/20	20/20	20/20	20/20	20/20	20/20	20/20	20/20

**Table 5 viruses-15-01484-t005:** Lytic activity of phages active against *Y. pestis* against set two of the *Yersinia* strains (collection of IRBA, France).

Bacterial Strains	Source of Isolation	Total Number of Strains	Susceptible to Phage Active against *Y. pestis*							
			YpEc57	YpEc58	YpEc2	YpEc11	YpYeO9	YpEc1	YpEc56	YpEc56D
*Y. pestis*	Human isolates	28	28/28	28/28	28/28	28/28	28/28	28/28	28/28	28/28
	Guinea pig	1	1/1	1/1	1/1	1/1	1/1	1/1	1/1	1/1
	not indicated	3	3/3	3/3	3/3	3/3	3/3	3/3	3/3	3/3
*Y. pseudotuberculosis*	Human isolates	2	1/2	1/2	1/2	1/2	1/2	1/2	1/2	0/2
*Y. frederiksenii*	Human isolates	1	0/1	0/1	0/1	0/1	0/1	0/1	0/1	0/1

**Table 6 viruses-15-01484-t006:** Lysis stability of phages active against *Y. pestis*—YpEcO9 and YpEc11—in liquid culture on *Y. pestis* EV76 (according to the method of Appelmans).

Lysis Stability	*Y. pestis* Phage	MOI of *Y. pestis* Phage and *Y. pestis* EV76								*Y. pestis* EV76
		1000	100	10	1	0.1	0.01	0.001	0.0001	
24 h	YpYeO9	-	-	-	-	-	-	-	-	0.5 MFTS *
	YpEc11	-	-	-	-	-	-	-	-	
48 h	YpYeO9	-	-	-	-	-	-	-	1 MFTS *	2 MFTS *
	YpEc11	-	-	-	-	-	-	-	-	
1 week	YpYeO9	-	-	-	-	-	-	-	2 MFTS *	3 MFTS *
	YpEc11	-	-	-	-	-	-	-	-	

* McFarland turbidity standard (MFTS). Regarding MFTS: 0.5 MFTS corresponds to 1.5 × 10^8^ CFU/mL, 1 MFTS corresponds to 3 × 10^8^ CFU/mL, 2 MFTS corresponds to 6 × 10^8^ CFU/mL, and 3 MFTS corresponds to 9 × 10^8^ CFU/mL. Average results of three parallel experiments are provided.

**Table 7 viruses-15-01484-t007:** Survival of *Y. pestis* phages YpYeO9 and YpEc11 in high ionic strength conditions and after transfer in distilled water. The results are the averages of three parallel experiments.

Phage	Viable Phage Counts (PFU/mL)		
	Initial Suspension	30 min after Exposure to 3.5 M NaCl	After Rapid Transfer to Distilled Water *
YpYeO9	1 × 10 ^7^	8 × 10 ^3^	8 × 10 ^3^
YpEc11	2 × 10 ^7^	5 × 10 ^4^	5 × 10 ^4^

* The titer calculated considering a 100-fold dilution of phages in distilled water (see Section 2.7).

## Data Availability

The data presented in this study are available in the article and in Appendix A.

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
