# Peer review of "New Bacteriophages with Podoviridal Morphotypes Active against Yersinia pestis: Characterization and Application Potential"

_viruses, 2023, doi:10.3390/v15071484_

Round 1
Reviewer 1 Report
In this work Suladze et al. isolated eight phages infecting Y. pestis from environmental samples in Georgia and characterized two of them (YpYe09 and YpEc11) in detail. They show that the phages exhibited a podoviridal morphology and that YpYe09 and YpEc11 are related to T7-like phages. In addition, the authors determined the stability of both phages under different conditions and investigated some lytic properties including the formation of phage resistant mutants.
The manuscript is well written and the experiments have been certainly performed carefully. The main problem that I have with this manuscript is that it is not clear to me, which data are really new and interesting. This is mainly because the authors do not compare their data thoroughly with those already published by others. This particularly pertains to the discussion section, but also the introduction. There are already a number of reports describing very similar Y. pestis phages exhibiting the same morphology, a very similar host specificity (including other Enterobacteriacea) and properties important for applications. Remarkably, these papers are all cited by the authors, but not discussed in detail. Therefore, this should be done to get a better impression, what is new or special with YpYe09 and YpEc11.
Some more specific comments:
1. Page 1, title of the manuscript: should be rephrased. “The Enterobacteriaceae-specific phages”, which? Are the phages specific for the whole family? Is it certain that they do not lyse members of another family? And is Y. pestis not also a member of this family?
2. Page 4, line 191: The lysis intensity should be described here in detail (see also points 7 and 9).
3. Page 7, lines 342 to 345: Please explain how eight different phages were isolated from six primary isolates.
4. Page 7, line 353: What is meant by “Negative Colony”?
5. Page 7, line 354: Seven or eight?
6. Page 8, line 373: The headline should be rephrased, e.g. by “Lysis of other species by the isolated Y. pestis phages”.
7. Page 9, Table 2: Please indicate the intensity of lysis, e.g. the efficiency of plating.
8. Page 10, Line 412: Please rephrase the headline, e.g. by “Host range determination for Y. pestis”.
9. Page 10, Table 4: Please give information on the biotype of the tested strains as well on the intensity of lysis.
10. Page 11, line 424: Please specify what is meant by “CL, SCL or OL”.
11. Page 18, lines 629 to 649: This section is a clear weakness of the manuscript. The authors tested the lytic activity of YpYe09 and YpEc11 on one strain (EV76) at one temperature (28 °C) using MOIs of 100 and 1.000. What about the activity at different temperatures and different MOIs? It is no wonder that the strain was efficiently lysed after few hours, perhaps by lysis from without. However, such high MOIs are not feasible under real conditions. Thus, this study, which is very important to assess the suitability of the phages for applications should be extended.
12. Page 20, line 680: What is meant by “negative plague”?
13. Page 20, line 700: Why can a similar activity be expected?
14. Page 21, line 741: This statement is not true, see e.g. Vagima et al., 2022 (reference 25).
Author Response
Dear Reviewer,
For responses, please, see the attached file

Reviewer 2 Report
The current manuscript describes screening and isolation process aimed to identify novel lytic phages specific to the highly lethal pathogen, plague causing bacteria, Yersinia pestis. As Y. pestis is not a common bacterium in the environment, and thus Y. pestis-specific phages are rare, the authors performed a two-step enrichment-isolation process. This process included phage enrichment in several other Enterobacteriaceae bacterial hosts, isolation of phages followed by selection against Y. pestis EV76 strain. Out of 75 environment samples, 54 phages specific to their primary host were found from which 8 phages were able to lyse also Y. pestis.
The 8 different novel phages were characterized for their morphology, host range, and lysing activity against 69 Y. pestis strains of different origins, revealing broad range phages, capable of lysing all tested strains. Two phages were further characterized for their potential as biocontrol agents, using various methods. Genome sequencing and annotation revealed no lysogenic genes and lysis stability and low mutation frequency rate confirmed their identity as virulent phages. The authors also determined growth cycle parameters using “one step assay”, using both Y. pestis strain and the enterobacterial strain that serve for their primary isolation. Latent time and burst size in Y. pestis were prolonged and lower, respectively, however they still posed sufficient values enabling their usage for biocontrol. Additional analysis revealed that both phages were stable in various solutions and physiological conditions, strengthening their potential application as anti Y. pestis biocontrol agents.
The manuscript is well written and the narrative is easy to follow, in general the major conclusions are supported by the data, and the experimental approaches employed are appropriate for the scientific issue addressed. There are some minor comments as followed:
Minor comments:
1- Line 25: I would suggest to mention that: “These phages lysed all 68 strains from 2 different sets of Y. pestis isolates…”
2- Line 31: “Bacterial challenge assay in liquid microcosm with YpYeO9/ YpEc11
phage mixture showed elimination of Y. pestis EV76 during 4 hours at P/B ratio 1000:1”
Q- Is there no "lysis from w/o" at such a high MOI? Please refer to this point in the result section #3.4.5.
3- Line 45: Please add the 4th recently recognized Y. pestis biovar: Microtus (Zhou et al, 2004).
4- Line 68: Ref # 16 is the same as Ref #6.
5- Lines 105-106: Please add ref.
6- Line 123: Please add to section 2.2. the biosafety level that used.
7- Line 141: Please mention growth temperature of the various strains.
8- Line 302: Please explain in a sentence or cite a reference describing the “liquid microcosm” you used. Is it different from regular flask?
9- Lines 307-308: Sample time should be the same as is shown in Figure 12. Additionally, did you determine the number of infective phage particles as mention? The results in Figure 12 are of bacterial count and not phage (as explained in line 646).
1- Line 419: Table 4 is unnecessary as the results are detailed in Table S6. The author can summarize the results in 1-2 sentences or alternatively summarize the results in a table that is grouped by Y. pestis source.
In addition, in Table S6 all 8 phages lysed all 36 tested Y. pestis strains (as also mentioned in line 423) while in Table 4 there are 2 bacterial strains that were not lysed. Please correct.
1- Table 5 states that 24/29 Y. pestis strains are susceptible to the 8 phages. This contradicts Table S7 (where all Y. pestis strains were lysed) and what is written in lines 431-432.
1- Line 434: “All phages but one (YpEc56D) were active against the strain of Y. pseudotuberculosis”
As there are only 2 Y. pseudotuberculosis strains, in which one is resistant and the other is susceptible to the phages, I would suggest: “All phages but one (YpEc56D) were active against 1 out of the 2 Y. pseudotuberculosis tested strains”
1- Figure 2: Please improve resolution.
1- Line 467: Figure 3: phage L-413C and not L-412C
15- Line 629: Please explain in paragraph 3.4.5 the logic of using a poor substrate media and a high MOI to characterize antibacterial efficacy in liquid.
1- Line 632: “In case of P/B ratio 100:1 (Figure 12a) gradual reduction in bacterial counts was observed during the first 4 hours”.
Thus, in Figure 12- Y axis title should be CFU/ml and not PFU/ml and Figure 12 legend title is not “propagation of phage…”
1- Line 627: Should be Figure 11 and not Figure 13.
1- Line 772: Please correct “adsorption efficacy” to “burst size”.
1- Table 2: Please check Table 2 in relation to the results shown in Tables S4 and S5. There are some differences.
2- Table 3: Phage group #5 shows the same typing as group #2 and thus can be unified.
2- Supplementary Tables: Please keep phage order in Tables S5, S6 and S7 the same as in Table 2.
2- Tables S1-S2: Is the biovar of the Y. pestis strains or some of them known? If known, please add this information and mention if you have representatives from the various biovars?

Author Response

(The authors gave the same response as above.)

Round 2
Reviewer 1 Report
For practical reasons I have given my responses point by point to the response of the authors.
Point 1: Page 1, title of the manuscript: should be rephrased. “The Enterobacteriaceae-specific phages”, which? Are the phages specific for the whole family? Is it certain that they do not lyse members of another family? And is Y. pestis not also a member of this family?
Response 1: Yes, we do agree that the title of the manuscript “the Enterobacteriaceae – specific bacteriophages with Podoviridae morphotype show lytic activity to Yersinia pestis: characterization and application potential” doesn’t fully and cleraly reflect charctersistics and origin of these phages and may lead to a confusion (also because Y. pestis is a member of the same family). We know that our new phages are active against some genera of Enterobacteriaceae (Eshcerichia, Shigella, Salmonella, Yersinia) but we do not have the data to say the same for other genera (such as Klebisella, Proteus, Serratia etc). So, we think the title can be changed following way: “New bacteriophages with Podoviridae morphotype active to Yersinia pestis: characterization and application potential” (Please, see lines 2-3-4 in the current working version of the manuscript).
My response: O.K., but use the word “podoviridal” instead of “Podoviridae” because the taxonomy of tailed phages has changed
Point 2: Page 4, line 191: The lysis intensity should be described here in detail (see also points 7 and 9).
Response 2: On the page 4, line 191, the intensity of lysis (in the phage spot test for determination of the host range) hasn’t been described in more details because we refer to the section 2.4. where (page 4, line 173-175) following detailed explanation has been provided: “The results were assessed by development of lytic reaction zones registered as: CL, complete lysis; SCL, semi confluent lysis; OL, overgrown lysis, presence of single bacterial colonies on spot; IPO/IPC, multiple opaque or clear phage plaques; R, resistant. But if you think that it still might be necessary to repeat this detailed descripotion in the section 2.6.2., we will follow your advice. As to points 7 and 9 we will provide
the response separately.
My response: I think that it would really make sense to give some quantitative data (EOP) at least on Y. pestis, because the description stated above is more qualitative than quantitative.
Point 3: Page 7, lines 342 to 345: Please explain how eight different phages were isolated from six primary isolates.
Response 3: The natural primary phage lysates (obtained from enriched samples) quite often are mixtures that may result on next steps (cloning/ purification) in the isolation of more than one phage from one primary phage lysate and those phages can differ by a number of properties. On the page 7, line 358 it’s mentioned that there were “6 primary phage isolates (some of them in mixutres)”. Specifically, out of this 6 primary phage lysates 2 were mixtures and resulted in 2 phages each: YpEc1 and YpEc2, and YpEc56 and YpEc56D..
My response: O.K.
Point 4: Page 7, line 353: What is meant by “Negative Colony”?
Response 4: We understand that the term “negative colony” or “negative plaque” might sound confusing. But those, especially “negative colonies” have been the terms commonly used in previous decades in the phage studies and phage producton at Eliava Institute , also used by some authors now as well. Under the “negative colony” we mean a zone of lysis or inhibition (clear ot translucent) that is formed by infectious phage particle on a lawn of bacteria on the solid medium.
The same meaning as for the term “phage plaque” (Kropinski et al, 2009). Considering your concern, we removed this word “negative” and used only “phage plaques “(Please, see the line 333, 373 and also 710).
My response: O.K.
Point 5: Page 7, line 354: Seven or eight?
Response 5: Of course, we are talking about 8 phage isolates in total. Here the number 7 applies to seven phages which formed phage plaques of the same size and morphology (4 – 6 mm in diameter, clear center and narrow turbid hallow zone), while one phage (YpEc2) produced smaller size plaques (1.5-2 mm). To make this clearer, we made change in the paper text (see lines 375 -376)
My response: O.K.
Point 6: Page 8, line 373: The headline should be rephrased, e.g. by “Lysis of other species by the isolated Y. pestis phage’
Response 6: Considering your suggestion we have rephrased the title of the subsection 3.2.2. as follows: “Lysis of different species of Enterobacteriaceae by newly isolated Y.pestis phages”. (Please, see the lines 395 -396)
My response: O.K.
Point 7: Page 9, Table 2: Please indicate the intensity of lysis, e.g. the efficiency of plating.
Response 7: In the Table 2 (as well as in the similar Tables 4 and Table 5) we provided the results of the host range studies by showing the number of strains susceptible to the studied phages out of the total number of strains of that species used. The format of such tabels doesn’t allow to specify the intensity of lysis or efficiency of plating on a particular strain. Thus, those data were not presented
here. We can just note that in the host range studies we considered results as positive only when intensity of lytic reactions in the spot test was registered as following: CL, complete lysis; SCL, semi confluent lysis; OL, overgrown lysis, presence of single bacterial colonies on spot; IP, multiple individual phage plaques. The detailed lysis intensity can be seen in corresponding supplementary
Tables S4 and Table S5 - for these tables indication/reference is given in the paper text (see lines 404– 410). As to tables 4 and 5, their supplementary Tables are S6 and S7 and corresponding indication we have been placed in the paper text (see lines 444 - 447 and 454 - 460). As to efficiency of plating (EOP), we determined this for randomly selected strains using serially diluted phages calculated by dividing the titer of the phage on the test strain by the titer of the same phage on its host strain. The results were comparable with the results of screening (phage spot test): e.g. CL type of lytic reaction was correspondng to a high EOP (0.5 - 1.0), OL and SCl - to a medium EOP (0.1- 0.5); and IP- to a low EOP (0.001-0. 1). Those data are not presented here.
My response: O.K., but please also consider my response to point 2.
Point 8: Page 10, Line 412: Please rephrase the headline, e.g. by “Host range determination for Y. pestis”.
Responce 8: We think that suggested headline not fully describes the content of subsection 3.2.3. - not clear host range of what is determined (of Y. pestis phages, right?). So, if you don’t mind, we 3 would prefer to keep the existing headline “Lytic Activity of Y. pestis Bacteriophages against Y. pestis Strains.
My response. I am honestly not happy with this headline, because it is misleading to talk about the lytic activity of Y. pestis phages against Y. pestis strains and because other Yersinia species were also examined in this chapter.
Point 9: Page 10, Table 4: Please give information on the biotype of the tested strains as well on the intensity of lysis.
Responce 9: We were not able to get such information regarding IRBA and NCDC Y. pestis strain collections. Probably biotyping of strains was not done.
As to intensity of lysis we have addressed this issue in the answer for the point 7.
My response: O.K., but please also consider my response to point 2.
Point 10: Page 11, line 424: Please specify what is meant by “CL, SCL or OL”.
Responce 10: Answer to this comment has been provided in the response 7: CL, SCL or OL describe the intensity of lytic reaction obtianed by phage spot test on a bacterial lawn on a solid medium. CL type (complete/fully clear lysis) indicates high intensity of lytic reaction; types OL (overgrown lysis, with presence of single bacterial colonies on spot) and SCL (semi confluent lysis) correspond to considerably less, but still sufficient to high intensity of lytic reaction.
My response: O.K.
Point 11: Page 18, lines 629 to 649: This section is a clear weakness of the manuscript. The authors tested the lytic activity of YpYeO9 and YpEc11 on one strain (EV76) at one temperature (28 °C) using MOIs of 100 and 1.000. What about the activity at different temperatures and different MOIs? It is no wonder that the strain was efficiently lysed after few hours, perhaps by lysis from without.
However, such high MOIs are not feasible under real conditions. Thus, this study, which is very important to assess the suitability of the phages for applications should be extended.
Responce 11: We were sorry to read about the dissapointment of the reviewer regarding the experimental series presented in the section 3.4.5. Let us to provide some explanations: these experiments (model study) in the liquid microcosms were done to show the potential of application of new phages not for therapeutic purposes but as a mean for removal of Y. pestis bacteria from enviroment, particurly water-containing environment (e.g. assuming storage tanks for water).
That’s why for this investigation the optimal temperature for Y. pestis - 28°C was chosen and not other temperatures, including human body temperature (37°C). Since such treatment aims rapid removal of contaminating bacteria, we decided to focus on the high MOI’s: 100 and 1000. Such MOIs have been shown to be effective in other studies, e.g for removal of Salmonella in liquid system (Turki et al, 2012) (also in our studies with other bacteria and phages) for bacterial removal
from contaminated surfaces. Yes, in the conditions of high MOIs there’s a posiibility to induce lysis from without (LO) although as Abedon mentioned (Abedon, 2011) “LO represents not just lysis but premature lysis, the rapidity of that lysis can be < 5min”. So, we think that lysis that occurs by combined action of phages YpEc11 and YpYeO9 in 4hrs can’t be considered just as LO (but such
possibility exists, of course), also we observed maintenance and even propagation of phages under high MOI (data not shown here). At the same time, of course, we agree with you that such high MOIs are not seen in the nature and
also not used in phage therapy applications, so definitely there’s a need to extend the studies with other temperatures and P/B ratios (including investigation of possible LO) to asses the suitability of the phages for other applications (as you mentioned in your comment). To address your concerns we added a sentence in the text “discussion” section (see lines 844 – 852).
My response: I am sorry but these explanations are not really plausible. Where do the authors know how many Y. pestis cells are in a water tank? And how do they want to manage to apply enough phages, if the tank is highly contaminated? And why 28 degrees and not lower temperatures? In my opinion, it would make much more sense to use significant lower MOIs (e.g. 0.1 or 0.01) and a temperature of e.g. 20 degrees to demonstrate the suitability of the phages for such an application.
Point 12: Page 20, line 680: What is meant by “negative plague”?
Response 12: We mean a zone of lysis or inhibition (clear ot translucent) that is formed by infectious phage particle on a lawn of bacteria on the solid medium. As we agreed above (see point 4) further the term “ phage plaque” will be used. Relevant change has been done in the text (see line 710)
My response: O.K.
Point 13: Page 20, line 700: Why can a similar activity be expected?
Response 13: Good question: yes we can’t expect phage activity here similar to the results of Skurnik et al (2021). We can just rephrase this sentence as following: “According to other studies (78) some Y. pestis phages can be strictly specific to Y. pestis, while host range of other phages included also Escherichia coli and Yersinia pseudotuberculosis”. The corresponding chnge has been done (line 730 - 732).
My response: O.K.
Point 14: P age 21, line 741: This statement is not true, see e.g. Vagima et al., 2022 (reference 25).
Response 14: Yes, you are correct especially if talking about phage therapy applications in experimental animal studies. The statement in the paper text (page 21 line 741) ”Till now the majority of the existing Y. pestis phages are routinely used for diagnosis of plague, but there are no available data on phage application as therapeutic agents”- applies to the phage application for patient’s/infected human’s treatment that currently hasn’t been in practice (at least accrding to the
available scinetific literature). Based on your commnet, we made following changes/additions to the text that describes in brief the studeis done by Vagima et al: “Till now the majority of existing Y. pestis phages are routinely used for diagnosis of plague. As to phage application as therapeutic agents for Y. pestis infection, a number of experimental animal studies have been done. Vagima et al (2022) have used mouse model for pneumonic plaque to assess the phage therapy potenial using known phages Ñ„A1122 and PST. Phage application
significantly delayed mortality and limited bacterial proliferation in the lungs but couldn’t prevent bacteriemia. To compensate certain insufficiency in treatment, the combination of these phages with antibiotic ceftriaxon was used that lead to survival of all infectd animals, thus demonstrating synergistic protective effect” (Please, see lines 771 - 779)
My response: O.K.
Author Response
Please, see the attachment
